# Degradation Behavior and Mechanical Integrity of a Mg-0.7Zn-0.6Ca (wt.%) Alloy: Effect of Grain Sizes and Crystallographic Texture

**DOI:** 10.3390/ma15093142

**Published:** 2022-04-26

**Authors:** Benjamin Millán-Ramos, Daniela Morquecho-Marín, Phaedra Silva-Bermudez, David Ramírez-Ortega, Osmary Depablos-Rivera, Julieta García-López, Mariana Fernández-Lizárraga, Argelia Almaguer-Flores, José Victoria-Hernández, Dietmar Letzig, Sandra E. Rodil

**Affiliations:** 1Instituto de Investigaciones en Materiales, Universidad Nacional Autónoma de México, Mexico City 04510, Mexico; divadql@gmail.com (D.R.-O.); odepablos@quimica.unam.mx (O.D.-R.); srodil@unam.mx (S.E.R.); 2Posgrado en Ciencia e Ingeniería de Materiales, Universidad Nacional Autónoma de México, Mexico City 04510, Mexico; 3Unidad de Ingeniería de Tejidos, Terapia Celular y Medicina Regenerativa, Instituto Nacional de Rehabilitación Luis Guillermo Ibarra Ibarra, Mexico City 14389, Mexico; cdmorquechodaniela@hotmail.com (D.M.-M.); phaedrasilva@yahoo.com (P.S.-B.); garcia.juli2013@gmail.com (J.G.-L.); fl.marianac23@gmail.com (M.F.-L.); 4Posgrado en Ciencias Médicas, Odontológicas y de la Salud, Ciencias Odontológicas, Universidad Nacional Autónoma de México, Mexico City 14389, Mexico; 5Departamento de Ingeniería Metalúrgica, Facultad de Química, Universidad Nacional Autónoma de México, Mexico City 04510, Mexico; 6Posgrado de Doctorado en Ciencias en Biomedicina y Biotecnología Molecular, Escuela Nacional de Ciencias Biológicas, Instituto Politécnico Nacional, Mexico City 11340, Mexico; 7Laboratorio de Biointerfaces, Facultad de Odontología, Universidad Nacional Autónoma de México, Mexico City 04510, Mexico; argelia.almaguer@mac.com; 8Institute of Material and Process Design, Helmholtz-Zentrum Hereon, 21502 Geesthacht, Germany; dietmar.letzig@hereon.de

**Keywords:** magnesium, biodegradable, annealing, mechanical properties

## Abstract

The microstructural characteristics of biodegradable Mg alloys determine their performance and appropriateness for orthopedic fixation applications. In this work, the effect of the annealing treatment of a Mg-0.7Zn-0.6Ca (ZX11) alloy on the mechanical integrity, corrosive behavior, and biocompatibility-osteoinduction was studied considering two annealing temperatures, 350 and 450 °C. The microstructure showed a recrystallized structure, with a lower number of precipitates, grain size, and stronger basal texture for the ZX11-350 condition than the ZX11-450. The characteristics mentioned above induce a higher long-term degradation rate for the ZX11-450 than the ZX11-350 on days 7th and 15th of immersion. In consequence, the mechanical integrity changes within this period. The increased degradation rate of the ZX11-450 condition reduces 40% the elongation at failure, in contrast with the 16% reduction for the ZX11-350 condition. After that period, the mechanical integrity remained unchanged. No cytotoxic effects were observed for both treatments and significant differentiation of mesenchymal stem cells into the osteoblast phenotype was observed.

## 1. Introduction

Mg-based materials have gained increasing attention for the fabrication of temporary implants in craniofacial and orthopedic applications [1,2,3,4,5]. Mg alloys should possess, and retain for a particular time, sufficient mechanical integrity to assist the healing of bone defects. Moreover, it should have appropriate degradability over time to avoid the potential adverse effects of secondary surgeries needed to remove the non-resorbable implants after bone healing [6].

Several studies show encouraging results of different Mg alloys (e.g., WE43, ZX alloys, etc.) used for the fabrication of biodegradable implants [1,7,8,9,10]. Different alloys have been developed in order to reduce the high degradation rates and alkalization of the surrounding biological media that induces cytotoxicity [11], the rise of ionic concentrations, and the release of insoluble products. Another concern, at an early stage of implantation, is that high gas (H_2_) production and accumulation due to the biocorrosion process also induces adverse effects, such as skin necrosis and osteolytic lesions. Furthermore, the high corrosion rates of Mg alloys as biodegradable implants may induce the loss of their mechanical integrity before the time-lapse needed for bone tissue regeneration [12,13].

Among the different options, the MgZnCa alloys have the advantage of containing only biocompatible elements and in vivo studies reported the excellent potential for osteosynthesis implant applications [11,14,15]. However, some drawbacks persist, Yu Y. et al. [16] reported that high purity Mg screws implants increased the pH values of the peri-implant bone marrow at four weeks after implantation in rabbits, where pH values significantly increased compared to Ctrl+ (no Mg screws). Grillo et al. developed a novel ZEK100 Mg alloy with a reduced amount of rare earth elements showing a significantly lower degradation rate in comparison with Mg in a complete cell culture medium [4]. They observed that the accumulation of corrosion products and pH change were higher in the case of Mg compared to ZEK100. In a clinical trial, Kim et al. reported promising results for Mg-5Ca alloy screws, although two cases of dehiscence related to gas accumulation were observed [17]. In the same way, Holweg et al. reported excellent results for Mg-0.45Zn-0.45Ca alloy screws for ankle fractures fixation but also observed an increased amount of gas accumulation in a patient attributed to a feasible osteoporotic condition [15].

The development of an ideal MgZnCa alloy must consider an optimal selection of the alloy composition and microstructure, i.e., grain size, amount, and distribution of precipitates and the crystallographic textures, which can be tailored by a proper selection of the thermomechanical treatment. These microstructural features can greatly affect the mechanical and corrosion resistance properties of the Mg alloys [1,18,19]. Therefore, the understanding of the relationship of microstructural features with corrosion properties is of high interest to develop optimum processing and production routes. On this matter, it has been reported that fine-grained microstructures with a well-balanced amount of precipitates offer a good combination of mechanical properties and ductility [20,21,22]. Meanwhile, adding crystallographic texture can reduce the corrosion/degradation rate [23]. In this regard, Hou et al. [6] reported the degradability, mechanical integrity, and in vitro cytocompatibility of a twin roll Mg-0.7Zn-0.6Ca (ZX11) alloy finding noticeable differences in the degradation rate and mechanical integrity between the hot-rolled ZX11 alloy and after recrystallization annealing at 400 °C. The differences were mainly attributed to deformation twins and residual strain stored in the as-rolled condition, which seemed to promote localized degradation, leading to a relatively fast deterioration in the mechanical properties. Meanwhile, the annealed condition showed relatively lower mechanical strength yet a lower degradation rate and a stable ductility during the initial weeks of immersion in modified Eagle medium alpha with 10% Fetal Bovine Serum (FBS) under cell culture conditions. These findings suggest that the same alloy can degrade differently depending on the internal defects (i.e., deformation twins and residual stress). However, the selection of the annealing temperature at 400 °C (below the rolling temperature of 450 °C) was not thoroughly explained. This work aims to evaluate the effect of the annealing recrystallization temperature (350 vs. 450 °C) of the ZX11 alloy on the degradation rate, mechanical integrity, cytocompatibility-differentiation of mesenchymal stem cells, and the antibacterial response. Thus, providing an overview of the material’s response produced with a different processing route.

## 2. Materials and Methods

### 2.1. Materials

Mg-0.7Zn-0.6Ca (wt.%) alloy strips with 5.2 mm thickness produced by twin-roll casting (TRC) were used for hot rolling experiments. A Novelis™ Jumbo 3CM twin-roll caster (TRC, Novelis, Voreppe, France) at Helmholtz-Zentrum Hereon (Geesthacht, Germany) was used to produce the strips. Before rolling, the strip was homogenized at 450 °C for 16 h. Hot rolling was performed at 370 °C. The rolling schedule consisted of three initial rolling passes with φ=0.1 and seven passes with φ=0.2:(1)$$\varphi =-\ln(\frac{h_{n+1}}{h_{n}})$$
where φ is the true strain, n is the number of the passes, and $h_{n}$ the sample thickness after the nth pass. After each rolling step, the sheet was reheated to the rolling temperature and held there for 10 min. After the last rolling pass, the samples were air-cooled.

Recrystallization annealing was performed at two different temperatures to promote differences in the grain size, amount, and distribution of precipitates and crystallographic textures. A set of samples was annealed at 350 °C for 30 min (hereafter ZX11-350 condition) and a second set was annealed at 450 °C for 30 min (hereafter ZX11-450 condition). Annealing temperatures above 250 °C were selected to ensure that the only secondary phase is the more anodic Mg_2_Ca phase, whose fraction can be changed according to the pseudo-binary phase diagram presented in Appendix A.

### 2.2. Microstructural, Structural and Composition Characterization

Metallographic examination of the Mg-0.7Zn-0.6Ca alloy for both conditions was conducted by optical microscopy and scanning electron microscopy (SEM). The samples were ground with emery SiC paper to grit 2500 (Schmitz-Metallographie GmbH, Herzogenrath, Germany) and polished with water-free silicon oxide suspension (OPS 0.5 µm) for optical microscopy. In the case of electron backscatter diffraction characterization (EBSD), the samples were electropolished using a Struers™ AC2 solution at 16 V for 35 s at −25 °C. EBSD measurements were carried out using a field emission gun scanning electron microscope Zeiss™ Ultra 55 equipped with an EDAX-TSL OIM™ system (Version 7.1.1x64, Ametek-EDAX Inc, Mahwah, NJ, USA). EBSD measurements were performed on the transversal direction (TD) and rolling direction (RD) planes. Energy-dispersive X-ray spectroscopy (EDS) was performed on mechanically polished samples to reveal the type and chemical composition of the secondary phase particles for both conditions. The fraction area of secondary phase particles was determined by using the computer-aided program ImageJ 1.53e (NIH-LOCI, University of Wisconsin, Maddison, WI, USA).

The global texture was measured using a Panalytical™ X-ray diffractometer (Malvern Panalytical, Almelo, Netherlands) with Cu-Kα radiation. The orientation distribution function (ODF) was calculated using the MTEX toolbox (Open source software 5.3.1, Freiberg, Germany) [24] from six measured pole figures $\left(0001\right)$, $\left\{10\bar{1}0\right\}$, $\left\{10\bar{1}1\right\}$, $\left\{10\bar{1}2\right\}$, $\left\{10\bar{1}3\right\}$ and $\left\{11\bar{2}0\right\}$. The results are presented in terms of the recalculated (0001) and {1010} pole figures.

The surface morphology of ZX11-350 and -450 samples, before and after immersion tests, was observed using a JEOL 7600F field emission-scanning electron microscope (FE-SEM, Tokyo, Japan). The composition of the corrosion products deposited on the sample’s surface was measured using EDS.

The crystalline structure of both conditions, before degradation assessment, was determined by X-ray diffraction (XRD) using a Rigaku Ultima IV diffractometer (RIGAKU, Tokio, Japan)with a Cu lamp. The measurements parameters were the following: a step of 0.02°, a scan speed of 1°/min and 2θ range from 3°–80°. Meanwhile, the degradation products’ structure was evaluated using the grazing incidence (0.5°) configuration. The degradation products identification was conducted using the PDXL2^®^ software (RIGAKU, Tokio, Japan) and selecting the lowest figure of merit on the different phases.

### 2.3. Degradation Evaluation

#### 2.3.1. Electrochemical Characterization

ZX11-350 and -450 alloy samples of 1 × 3 cm^2^ were ground with SiC emery paper up to grit 2000, cleaned with acetone, subsequently immersed for 15 min in isopropyl alcohol in an ultrasonic bath, and dried under airflow at room temperature.

A standard three-electrode cell connected to a Gamry Reference 600 potentiostat was used for the electrochemical assessment of the ZX11-350, and ZX11-450 samples at 37 °C in 150 mL of Dulbecco’s Modified Eagle’s Medium/Ham F-12 50/50 Mix (DMEM/F-12; Cat. 100092-CV, Corning^®^, New York, NY, USA) supplemented with 10% *v*/*v* Fetal Bovine Serum (FBS, Ct. 160000 44-PRO, Gibco^®^, New York, NY, USA), the exposed area was 1.13 cm^2^. A saturated Calomel electrode (SCE) and graphite rod were used as reference and counter electrodes, respectively. A more detailed description of the methods followed for assessing the electrochemical response of the samples can be reviewed in [25]. At least three samples were considered for each measurement. The corrosion current density was obtained with Tafel analysis of the cathodic branch and the degradation rate (*P_i_*) was determined according to ASTM G102-89 [26].

#### 2.3.2. Hydrogen Evolution Test

Hydrogen evolution of the ZX11-350 and ZX11-450 samples was measured in a closed system consisting of a 127 mL glass reactor coupled to a gas chromatograph. Before any measurement, N_2_ was introduced into the system to displace any residual O_2_ (chromatographic peak). Hydrogen evolution was measured every 20 min for 4 h using a gas chromatograph device (Shimadzu GC-2014, Kyoto, Japan) equipped with a Shimadzu Molecular Sieve Column CU7669-1 and a thermal conductivity detector. N_2_ was used as carrier gas. The samples were individually adhered to a quartz tube inside the reactor, exposing the frontal and lateral faces (exposed area 1.72 cm^2^), and subjected to continuous stirring at 500 rpm. The solution was 50 mL, and DMEM/F-12 supplemented with 10% *v/v* FBS and was maintained at 37 °C. The volume to area ratio (V/A) was set to 29 mL/cm^2^. Three measurements were performed for each condition. The amount of H_2_ evolved in terms of moles is determined according to Section 2.1.

#### 2.3.3. Immersion Test

ZX11-350 and ZX11-450 samples with dimensions of 10 × 10 × 1.8 mm^3^ were used for the immersion test. Samples were manually ground with SiC paper from #360 up to #2000 grade until a homogeneous but not mirror-finished surface was obtained. Finally, samples were immersed in Nital solution (3% *v*/*v* nitric acid alcoholic solution) for 10 s, rinsed with deionized water, dried with pressurized air, and stored in hermetic plastic bags.

Individual samples were weighed (Analytical balance Ohaus) to determine their initial weight (W_0_) and sterilized with UV light (UVC Cross Linker 500 Hoefer, Hollister, MA, USA ), positioned in an acrylonitrile butadiene styrene (ABS) 3D printed device specifically designed to allow homogeneous interaction of the samples’ surface with the incubation medium (DMEM/F-12 supplemented with 10% *v*/*v* FBS) under orbital shaking at 37 °C. The surface to medium volume relation was 0.40 mL/mm^2^ according to the ASTM G3172 (2004).

The sample’s weight was measured at different incubation intervals, 7, 15, 30, and 45 days. Independent samples were removed from the incubation medium, dried, and weighed at each time interval. Experiments were performed by quadruplicate; after drying, one sample was kept as obtained to characterize the degradation products on its surface, and three samples were immersed in a chromic oxide solution in distilled water (18 g/100 mL) for 20 min to remove the degradation products, rinsed with deionized water and 100% ethanol, dried overnight, and weighed again (*W_t_*). The following equation was used to determine the degradation rate (*P_W_*) of the samples in millimeters per year:(2)$$P_{W}=\left(87,600\;\frac{h\;mm}{cm\;year}\right)\times \frac{∆m}{A\rho_{Mg}t}$$
where ∆m corresponds to the mass loss (*W*_0_–*W_t_*) in grams, *A* is the surface area of the sample in cm^2^, *ρ_Mg_* corresponds to the alloy density considered as that of *Mg* (1.748 g/cm^3^), and *t* is the immersion time in hours [6].

#### 2.3.4. Magnesium Release

The samples were independently submerged in DMEM/F-12 supplemented with 10% *v*/*v* FBS (samples weight to medium volume ratio of 0.2 g/mL; according to the ISO 10993-12 standard) and incubated at 37 °C and 5% CO_2_ for different periods, 1, 3 and 7 days. At each measurement time, independent samples were removed from the incubation medium, and the amount of magnesium released into the medium was determined using the kit HI 93719-0 (Hanna Instruments^®^ Ltd., Leighton Buzzard, UK) and a portable photometer HI 96719 (Hanna Instruments^®^ Ltd., Leighton Buzzard, UK). The experiments were performed twice by duplicate per condition.

#### 2.3.5. pH Evolution Monitoring

Four samples of each condition were individually immersed in DMEM/F-12 supplemented with 10% *v*/*v* FBS, using a weight to medium volume ratio of 0.2 g/mL (ISO 10993-12 standard), and incubated at 37 °C and 5% CO_2_. The pH of the medium was monitored with a 340 Beckman pH Meter coupled to a micro pH electrode (Hanna instruments) at different time intervals, that is, immediately after contact with the alloy, and then, at 15 min, 30 min, 1 h, 3 h, 24 h, 48 h and 72 h of incubation.

### 2.4. Mechanical Integrity

The assessment of the mechanical properties after different periods of immersion (15, 30, and 45 days) in DMEM/F-12 supplemented with 10% FBS was performed using the tensile assay according to ASTM E8 [27]. Immersion conditions and samples surface to medium volume ratio were the same as those described in Section 2.5, but specimens were allocated in individual 3D-printed ABS holders specifically designed to allow homogeneous exposure of bone-shaped specimens to the immersion medium. Three samples for each period were considered for the tensile tests, carried out in a universal testing machine (Shimadzu AGS-X 10 kN, Jiangsu, China) at 0.1 mm/min testing speed (strain rate 1.6 × 10^−4^ s^−1^).

### 2.5. In Vitro Evaluation

Samples of the ZX11-350 and -450 conditions were prepared as described in Section 2.3.3 for the following in vitro biological evaluations.

#### 2.5.1. Cytotoxicity Assays

The potential cytotoxicity of ZX11-350 and -450 conditions was tested by culturing mesenchymal stem cells derived from human adipose tissue (MSC-Ad) in the presence of the sample’s lixiviate products (extracts) at different dilutions and using the MTT and LIVE/DEAD assays to characterize the viability of the cells after different periods of incubation with the extract’s dilutions. The MSC-Ad isolation and characterization methods are detailed in the Appendix A. MTT experiments were performed in quadruplicate, and LIVE/DEAD assays were performed in triplicate per condition.

#### 2.5.2. Differentiation Assays

Cell culture inserts were used to evaluate the differentiation of the MSC-Ad cells into the osteoblastic phenotype. Firstly, the cell viability was estimated since the cells were exposed to the degradation products without dilution, and then immunohistochemistry assays were used to obtain the osteoblasts markers. MSC-Ad were seeded (5000 cells/cm^2^) in 6-well tissue culture plates and cultured with DMEM/F-12 (Corning^®^, New York, NY, USA) supplemented with 10% *v*/*v* FBS and 1% anti-anti. After 24 h of culture, the medium was removed, sterilized samples were individually placed on standing cell culture inserts (Millicell^®^, Merck Millipore Ltd., Cork, Ireland) and placed on each cell culture (each culture well), the fresh culture medium was added, and culture plates were set back in the incubator. The culture medium was changed every 24 h. After 1, 3, and 6 days of culture, the hanging inserts with the alloys were removed, and cell viability was independently assessed from individual cell cultures by the MTT (Sigma Aldrich; St. Louis, MO, USA) and LIVE/DEAD assays (Invitrogen, Thermo Fisher Scientific, Eugene, OR, USA) using the same protocols and number of replicates as described above to characterize the viability of cells under incubation with the diluted extracts.

For immunocytochemistry assays, MSC-Ad were seeded in 6-wells culture plates at a density of 2 × 10^4^ cells/cm^2^ and incubated for 24 h at 37 °C and 5% CO_2_ with DMEM/F-12 supplemented with 10% *v*/*v* FBS and 1% anti-anti. Then, the culture medium was removed, alloys samples were individually placed on standing cell culture inserts on each cell culture, the fresh culture medium was added, and culture plates were set back in the incubator. Cell cultures were maintained for 8 days with the inserts and the ZX11-350 and ZX11-450 samples, refreshing the culture medium every 24 h. On incubation day 8, the inserts with the samples were removed, and cell cultures were washed with PBS, fixed with 2% paraformaldehyde solution, washed again with PBS, rinsed with distilled water, and incubated in 0.9% hydrogen peroxide for 10 min, protected from light. Subsequently, hydrogen peroxide was removed, and cells were rinsed with PBST (1X PBS—0.1% Tween 20) and incubated with blocking solution from VECTASTAIN^®^ Elite ABC-HRP Kit (Peroxidase, Universal; PK-6200, Vector Laboratories, Burlingame, CA, USA) for 30 min. Then, primary antibodies, mouse anti-human Runx-2 (Abcam ab 76956, 1:100 dilution), rabbit anti-human Osteopontin (Abcam ab8448; dilution 1:100), rabbit anti-human Osteocalcin (Santa Cruz FL 100 SC 30044; dilution 1:50), and mouse anti-human Collagen I (Abcam ab6308; dilution 1:1000), were independently added to cell cultures and incubated at 4 °C overnight. After incubation, cells were rinsed with PBST and incubated with VECTASTAIN^®^ Elite ABC-HRP Kit. Cell cultures were rinsed with PBST and visualized with 3,3-diaminobenzidine (DAB solution; DAKO liquid DAB, Cat. K3468) was added until the development of the immunocytochemistry staining. Finally, cells were washed with distilled water, counterstained with Meyer’s Hematoxylin, washed with water, and briefly rinsed with acidic alcohol (70% ethyl alcohol in hydrochloric acid, 0.5%). Micrographs were acquired in an Axiovert 25 HBO 50 microscope (Carl Zeiss, Jenna, Germany) at different magnifications (10X, 20X and 40X).

Semi-quantitative analysis of immunostaining results was performed using the Fiji ImageJ software. A color deconvolution was performed on the micrographs by selecting the *H&E* and *DAB* staining, brown and blue, respectively. Then, the brown-colored area was quantified (Positive Area), indicating the positive area. The total area covered by cells was also quantified. Quantifications were performed in triplicate for each condition and protein (Osteocalcin, Osteopontin, Collagen I, and Runx2). The percentage of positive protein expression was estimated using the following formula:(3)$$\mathrm{Positive}\;\mathrm{Expression}\;\left(\%\right)=\frac{\left(\mathrm{Positive}\;\mathrm{area}\right)*100}{\mathrm{Total}\;\mathrm{area}}$$

#### 2.5.3. Antibacterial Test

The effect of the lixiviated products of the ZX11-350 and ZX11-450 samples on the bacterial growth was determined by evaluating the planktonic growth of three aerobic reference strains associated with opportunistic and implantable device infections: *Escherichia coli* (ATCC 33780), *Staphylococcus aureus* (ATCC 25923), and *Staphylococcus epidermidis* (ATCC 14990).

A bacterial suspension with 10^5^ cells/mL of pure cultures of either *E. coli*, *S. aureus* or *S. epidermidis*, was added to the samples and incubated at 37 °C in an orbital shaker at 120 rpm under aerobic conditions for 1, 3, and 7 days. After incubation, the capability of the lixiviated products of the ZX11 samples to inhibit planktonic bacterial growth was estimated by counting the number of colony-forming units (CFUs) present in the incubation broth media taken from the inoculated ZX11 samples. All experiments were performed in triplicate. The antibacterial effect (inhibition ratio percentage) of the ZX11 samples against the bacterial strains tested was calculated according to the following equation:(4)$$\mathrm{Inhibition}\;\mathrm{ratio}\;\left(\%\right)=\left(\frac{B1-A1}{B1}\right)*100$$
where *A*1, is the number of colonies forming units (CFUs) obtained from the incubation media with the lixiviated products of the ZX11-350 or ZX11-450 samples, and *B*1 is the number of CFUs obtained from the incubation media without ZX11 samples (considering *B*1 as 100% of planktonic bacterial growth; negative control).

### 2.6. Statistical Analysis

For in vitro degradation evaluation and mechanical properties assessment, the statistical analysis was conducted considering the arithmetic average of at least three samples for each technique and condition; the error corresponds to the standard deviation.

Regarding the in vitro cytotoxicity and differentiation assays, statistical significance was determined by a two-way analysis of variance, followed by a Tukey’s multiple comparison test to compare pair groups. A value of *p* < 0.05 was considered statistically significant.

## 3. Results

### 3.1. Microstructural Characterization

Figure 1 compares the optical micrographs, SEM images, and the corresponding pole figures of the ZX11-350 and ZX11-450 samples. ZX11-350 (Figure 1a) shows a homogenously recrystallized microstructure with an average grain size (d) of 14 ± 6 µm and uniform distribution of precipitates inside the grains. The EDS (Table 1) analysis performed in the matrix (M1) and selected particles (1–4) confirm (Figure 1c) that the precipitates are Ca-rich (Mg_2_Ca), but Si and Fe impurities were present in some of them. Some fine precipitates sitting in a linear pattern within the grains were observed, agreeing with Hou et al., who observed that Mg_2_Ca particles tend to nucleate in twin boundaries [28], which after the activation of continuous static recrystallization are left undissolved inside the grains. The area fraction covered by these precipitates is ~0.48%. The microstructure of the ZX11-450 sample shows a coarse recrystallized microstructure with an average grain size of 45 ± 23 µm (Figure 1b). The precipitates are less numerous but slightly larger and showed a rich Ca content (1–4) than the matrix (M2). There is an apparent reduction of precipitates in the microstructure in which the area fraction is only 0.08%.

Moreover, the large, elongated particles (stringer-like) observed in the ZX11-450 sample contain a significant amount of oxygen. This reduction in the fraction of secondary phases is explained due to the larger solubility of Ca in the hcp solid solution at 450 °C, as predicted by the phase diagram.

Slight differences in the crystallographic orientation of the grains (i.e., crystallographic texture are shown in Figure 1e–f between both samples. Both samples presented basal textures. In the case of the ZX11-350 alloy, its texture is characterized by relatively strong peaks splitting from ND towards RD, and the $<10\bar{1}0>$ pole parallel to RD. In the case of the ZX11-450, a broad scatter of basal poles towards the transverse direction (TD) is also visible. However, the texture intensity of the ZX11-450 is significantly weaker than the ZX11-350 sample.

Further details of the microstructures are revealed using the kernel average misorientation (KAM) analysis, Figure 2. The KAM maps are calculated based on the degree of misorientation between a kernel (measuring point) and its neighbors [29]. Thus, in Figure 2a,b superimposed kernel average misorientation maps over image quality (IQ) maps are used to indicate zones with high stored strain. This is an important aspect of the microstructure as the ZX11-350 sample was recrystallized at a temperature lower than the deformation (rolling) temperature. It is also important to highlight that the KAM method only considers geometrically necessary dislocations (GND) for the estimation of stored energy and omits the statistically stored dislocations (SSD) [29]. Although KAM analysis might underestimate the strain, it indicates zones of higher stored strain. In Figure 2, it is clear that some grains in the ZX11-350 sample still contain a significant amount of GNDs (Figure 2a), whereas the ZX11-450 sample shows few indications of zones with high stored strain. The relatively high content of stored strain in the ZX11-350 sample can be related to incomplete recrystallization as the annealing temperature is below the rolling temperature. In contrast, in the ZX11-450 sample, the annealing temperature is above the recrystallization temperature leading to an almost strain-free condition.

### 3.2. Degradation Assessment

The degradation rate of the ZX11 samples was evaluated by different methods and immersion times. The short-term degradation (first hours of immersion) was evaluated by the standard electrochemical tests and hydrogen evolution in a closed reactor; meanwhile, the large-term degradation (days) was assessed by weight loss determination, Mg release, and pH variation. All the measurements (at least three replicates) were conducted in cell culture media supplemented with fetal bovine serum to simulate the physiological conditions closer.

#### 3.2.1. Electrochemical

Figure 3 shows that the electrochemical response is nearly identical among both samples despite the differences in grain size and the fraction of precipitates. The OCP variation with time (Figure 3a) shows that during the first 4000 s, there is an important modification of the surface, where the OCP oscillates, being slightly more anodic for the ZX11-350 sample at the first 3000 s, probably because the dissolution initiates in the more anodic Mg_2_Ca precipitates. However, later, the ZX11-450 becomes more anodic, and the trend remains unchanged after the EIS and LPR measurements.

EIS experiments were performed after stabilization of the OCP by 3 h. The Nyquist plots (Figure 3b) and Bode representations (Appendix A) are very similar between both samples, presenting three time constants that were fitted using the equivalent circuit presented in Appendix A. After averaging the values from three independent measurements, the polarization resistance was 1420 ± 117 Ωcm^2^ for ZX11-350 and 1500 ± 163 Ωcm^2^ for the ZX11-450. However, this difference is not statistically significant, i.e., from the short-term electrochemical evaluation, both samples are equivalent.

Linear polarization resistance (LPR) and potentiodynamic polarization measurements were conducted after the EIS, confirming this trend. Before each test, the OCP was measured for 10 min to guarantee no severe damage occurred on the samples between tests (Figure 3c). The slopes of the LPR plots indicated that the polarization resistance, which is inversely proportional to the corrosion rate, was 1562.5 ± 303.6 Ωcm^2^ and 1840 ± 371.9 Ωcm^2^ for ZX11-350 and ZX11-450, respectively, but again the difference is not statistically significant. Finally, the Tafel fitting of the PDP curves was used to determine the corrosion current density and degradation rate (*P_i_*) according to the ASTM G102-89 standard [26], obtaining 0.60 ± 0.06 and 0.53 ± 0.09 mm/year for the ZX11-350 and -450, respectively.

#### 3.2.2. Hydrogen Evolution

Since the degradation of Mg produces hydrogen, it is common to measure the amount of evolved hydrogen after days of immersion. Our approach is slightly different since we measure the short-term production of H_2_ using a chromatographer, which makes the analysis equivalent to the immersion times used in the electrochemical tests.

Figure 4 shows the evolution in time of hydrogen normalized to the exposed area (μmol/cm^2^) or equivalently in mg/cm^2^ (right axis). As it can be seen, ZX11-450 has slightly lower values, but the difference is not statistically significant when we take the average from three different samples. From the total amount of H_2_ evolved during the four hours of the test, the degradation rate (P_AH_) in mm/year can be estimated following Shi et al. [30], which assumes that each mol of H_2_ is equivalent to a mol of Mg. The values are slightly lower for the ZX11-450 samples (0.45 mm/year) than for the ZX11-350 (0.48 mm/year). These DR values were determined when FBS was added to the solution, but lower values were observed without FBS, as shown in Appendix A, remarking the importance of simulating the physiological conditions.

Figure 4b compares the short-term or instantaneous degradation rate calculated from PDP (P_i_) and H_2_ measurements (P_AH_), observing a good agreement between both values. The absolute values suggest a lower DR for the ZX11-450 samples, but indeed the difference between samples was not statistically significant. Interestingly, both P_i_ values are comparable to those reported for MgZnCa alloys with similar compositions, see Table 2. Moreover, the P_AH_ of both conditions is lower than the reported for Mg-5.3Zn-0.6Ca alloy, for which P_AH_ was also calculated with hydrogen evolution at short times [31]. Comparisons with other authors must be made carefully since the evaluation conditions (electrolyte, exposure time, etc.) may be different and strongly determine the corrosive response.

#### 3.2.3. Long Term DegradationImmersion Test

##### Immersion Test

Figure 5a plots the weight loss DR measured from the immersion test in supplemented cell culture medium. The long-term degradation rate (P_W_) on days 7 and 15 is ~15% higher for the ZX11-450 sample than the ZX11-350 sample, contrary to the trend observed by the electrochemical and HE data trend. However, both samples present no difference for longer immersion times.

As shown in Appendix A, no differences were obtained from the analysis of the corrosion products after the long-immersion tests between both samples. The products presented changes with immersion time from MgCO_3_·3H_2_O during the first days to Mg_5_(CO_3_)_4_(OH)_2_·4H_2_O and MgCl_2_ after 6 days.

#### 3.2.4. Mg Concentration

Figure 5b shows the Mg released from the alloys upon immersion into the culture media (DMEM/F-12 supplemented with 10% *v*/*v* FBS) from 1 to 6 days. Here, we observed a slightly higher fraction of Mg released from the ZX11-350 sample during the three periods. The Mg concentration observed for the ZX11-350 sample might be related to the more extensive dissolution of the Mg_2_Ca precipitates, which during the initial immersion suffer anodic corrosion. There is a more significant fraction of Mg_2_Ca precipitates in the ZX11-350 sample.

#### 3.2.5. pH Variation

From Figure 5c, it can be observed that the pH of the medium increased with immersion time for both conditions. From 0 to 24 h of immersion, the pH changed from 7.4 to ≈9. Later, at 48 h of immersion, the pH increment rate decreased compared to the first 24 h of immersion, probably exhibiting a slower corrosion rate. After 72 h of immersion, the pH significantly decreased due to the medium refreshing that was performed after 48 h of immersion, indicating that after the initial abrupt pH change upon the alloys’ contact with the medium, the alloys were passivated, and further contact with fresh medium did not longer seem to cause an abrupt reaction. The pH increment relates to the release of OH^−^ ions into the medium during the Mg corrosion process described before [40,41,42].

### 3.3. Mechanical Properties

The mechanical integrity of the ZX11-350 and ZX11-450 after 15, 30, and 45 days of immersion in DMEM/F-12 supplemented with 10% *v*/*v* FBS are presented in Figure 6. It can be appreciated a higher initial elongation at failure for ZX11-350 compared to the ZX11-450 sample, Figure 6a.

On the other hand, no differences are observed for initial ultimate tensile strength (UTS) between both samples in pristine conditions, and it is comparable with extruded Mg-0.5Zn-0.2Ca alloy [43] and rolled Mg-2.5Zn-0.2Ca alloy [44]. There is a slight decrease after day 15th for the ultimate tensile strength (UTS) of both conditions, but no significant changes are observed afterward (Figure 6b). It is most likely that UTS variations are due to the sudden specimen fracture at places with localized corrosion rather than to the weakening of the alloy.

The toughness is determined by integrating the area below the stress-strain curves, Appendix A, for both conditions and is presented in Figure 6d. The toughness is 17% higher for the ZX11-350 sample in comparison with the ZX11-450 sample at day 0 of immersion. After 15 days of immersion, this difference increased to 50%, with no further significant variations for more extended periods. These results are a straight consequence of the elongation at failure since no significant differences are observed for the YS and UTS for ZX11-350 and -450 after 15 days.

### 3.4. Biological Assessment

#### 3.4.1. Cytotoxicity, In Vitro Studies

Following the recommendations for Mg-based samples, the in vitro cytotoxicity must be evaluated using dilutions. Appendix A shows the MTT and LIVE/DEAD^TM^ assays results for the diluted (5X, 10X, and 15X dilutions, according to J Fischeer et al. [45]) samples lixiviates at different days of cell culture. In agreement with previous results reported by Hou et al. [6] for the ZX11 alloy, there is no cytotoxicity independently of the annealing treatment.

#### 3.4.2. Differentiation Assays

The osteoblast differentiation capability of the samples was evaluated using the immunocytochemistry assays, where the different osteogenic markers can be observed and quantified. Figure 7a shows the quantification of the nuclear expression of Runx-2, and the cytoplasmic expression of osteopontin, osteocalcin, and collagen I, as osteogenic differentiation markers, after 8 days of MSC culture in contact with the non-diluted lixiviates from ZX11-350 and ZX11-450. Figure 7b presents the images of the cells that were counterstained with hematoxylin and eosin and are observed in purple, while positive expression of osteogenic differentiation markers is observed in brown. Runx-2 and collagen I were chosen as early osteogenic differentiation markers. Runx-2 is essential in the initial stage of bone development and Collagen I is the most abundant protein in the extracellular matrix. In contrast, osteopontin and osteocalcin are late differentiation markers. Osteopontin and osteocalcin are two highly abundant non-collagenous proteins in the bone extracellular matrix synthesized by mature osteoblasts [46,47,48,49].

The images and the quantification indicate that the Ctrl (MSC cultured in fresh DMEM/F-12 supplemented with 10% *v*/*v* FBS and 1% anti-anti) do not significantly express the osteogenic markers. Contrary, the expression of these markers is promoted in the presence of the Mg corrosion products in agreement with previous reports that demonstrate that Mg promotes bone growth and regeneration [50,51,52,53,54].

#### 3.4.3. Antibacterial

The antibacterial effect of the ZX11-350 and-450 samples is presented in Table 3. The inhibition ratio varied depending on the bacterial strain; however, a total bacterial growth inhibition (100%) was achieved when bacteria were exposed to the ZX11-450 sample after 7 days of incubation, regardless of the strain tested.

The Gram-negative *E. coli* seems to be less sensitive to the effect of the lixiviated products of the ZX11alloy samples. Bacterial growth Inhibition of 47.5 ± 13.3 and 56.6 ± 13.2% was detected on day one of incubation in the presence of ZX11-350 and ZX11-450 samples, respectively. On day three of incubation, a more pronounced antibacterial effect was observed (93.02 ± 2.3 and 77.6 ± 4.3), however, at 7 days of incubation with the ZX11 alloy specimens, only the bacterial cells of *E. coli* that were exposed to the products of the ZX11-450 sample reached a total bacterial inhibition (100%).

On the other hand, the Gram-positive strains tested (*S. aureus* and *S. epidermidis*) were more susceptible to both ZX11samples, especially *S. aureus*. This strain was inhibited entirely (100%) after 3 and 7 days of incubation with either ZX11-350 or ZX11-450 samples. A slightly less susceptible to the lixiviated products of the ZX11 alloy samples was *S. epidermidis*. However, it seems that the corrosion/degradation products of the ZX11-450 sample were more effective in completely inhibiting the growth of this strain after 3 and 7 days of incubation.

## 4. Discussion

### 4.1. Degradation Rate

The different techniques used to evaluate the degradation rates of the ZX11-350 and ZX11-450 samples led to confusing results. The ZX11-350 has slightly larger corrosion, although not statistically different, according to the short-term data. However, long-term data indicated smaller corrosion but larger Mg ionic release. Explanation of these contradictory results comes from the differences in the microstructure since the composition of both samples is precisely the same. The dissimilar temperatures of the annealing-quenching treatment induced significant differences in the grain sizes, percentage of precipitates, and strain. According to the phase diagram and EDX analysis, the predominant secondary phases are Mg_2_Ca, which are known to be more anodic than the matrix [55], i.e., they will be corroded faster than the matrix at the initial immersion times. Once these anodic precipitates are consumed, degradation of the matrix will dominate the response.

The role of the grain size has been discussed in several works, reporting the beneficial/detrimental effects on corrosion resistance. Bahmani et al. [56] published a recent review article aiming to correlate corrosion rates with microstructural parameters of Mg-based alloys. They conclude that the recipe for lower corrosion rates is to use ultra fined-grains, alloying compounds with low volta-potentials and low fraction of secondary phases. Ultra-fine grains induce a lower degradation rate because the high energy of the grain boundaries accelerates the formation of the protective corrosion product layer. Ergo, the corrosion current density (and DR) of fine-grain materials is higher initially but decreases once the corrosion products protect their surface. According to this, we would expect that the ZX11-350 sample with a smaller grain size presented a larger electrochemical corrosion rate (Pi) and hydrogen evolution (P_AH_) than ZX11-450, but a lower P_w_ rate. The results indicated that actually, P_w_ is smaller, but not the short-term degradation rates, which are almost identical to the ZX11-450 sample. A possible explanation is the presence of a large number of Mg_2_Ca precipitates, which are anodic concerning the matrix, and therefore, degrade at a high rate, increasing the corrosion current density, hydrogen production, and Mg release. However, there is no significant weight loss since the matrix is anodically protected. However, once the anodic precipitates are dissolved, the corrosion propagates to the whole surface dissolving the matrix. Initially, the ZX11-350 produces a large corrosion current density (not expected for the smaller grain size) due to the dissolution of the precipitates, but once the precipitates are consumed, the large number of grain boundaries induce a fast covering with protective corrosion products, reflected in a lower DR at long-term (Figure 8).

On the other hand, the ZX11-450, with a larger grain size and lower number of precipitates, initiates the protective layer formation in the first hours of immersion, and the corrosion current density is produced predominantly by the matrix. Thus, the degradation rates after 30 days of immersion were similar for both annealing conditions. The weaker texture intensity of the ZX11-450 compared to than ZX11-350 sample might also contribute to the degradation. Song et al. observed that Mg grains with basal orientation show a more positive corrosion potential, larger impedance, and thinner surface film than the grains with non-basal orientation in 0.01 M NaCl solution [57]. Meanwhile, Liu et al. observed that grains with close (0001) orientation are the most corrosion-resistant in 0.1 N HCl [23].

The smaller grain size of the ZX11-350 in comparison with the ZX11-450 sample can improve corrosion resistance since it induces more homogeneous corrosion and a more coherent passivation layer. Therefore, the combined effect of a higher texture, precipitate’s composition and fraction, and smaller grain size of the ZX11-350 sample explain the lower DR observed at 7 and 15 days of immersion. However, the structural differences have compensated for each other at longer immersion times, leading to a similar DR of about 0.16 mm/year for both conditions after 45 days of immersion.

Our results are slightly different from Hou et al. [6], partially due to differences in the surface topography and the medium. Hou et al. measured the degradation rates using mirror-polished samples; meanwhile, we used samples ground up to 2000 SiC grade with an average roughness of 0.205 µm [25]. It has been shown before that roughness negatively affects the DR [58].

The dissolution of the precipitates in the ZX11-350 samples during the first days of immersion induces a more significant concentration of Mg released. However, a larger Mg release is not necessarily a detrimental factor, as long as it does not significantly disturb the osmotic balance of the surrounding biological environment [49]. Mg ions promote MSC osteogenic differentiation, enhancing bone formation and growth [59]. Juan et al. found that 1.2 mM to 1.8 mM MgCl_2_ concentration promotes MSC proliferation and facilitates differentiation towards the osteoblastic phenotype [57], which is within the range of Mg release from the present alloys, ZX11-350 and ZX11-450, in DMEM/F-12 supplemented with 10% *v/v* FBS; that is 10 mM (250 mg/L)—19 mM (475 mg/L).

### 4.2. Mechanical Integrity

We discuss first the differences in the mechanical properties between the ZX11-350 and ZX11-450 samples. The ZX11-350 samples presented larger ductility and toughness than ZX11-450, with values similar to those reported for the ZX11 sample annealed at 400 °C reported by Hou et al. [6]. This is correlated to the formability ability of Mg alloys, and it is influenced by several factors, such as texture, grain size, twinning and stored strain, among others. Commonly, for hexagonal close-packed systems, such as Mg and its alloys, there are two principal plastic deformation mechanisms at room temperature: dislocation slip (mainly basal <a> slip) and twinning. In particular, the intense texture with a preferred basal orientation towards RD for the ZX11-350, see Figure 1a, suggests basal <a> slip as an important deformation mechanism. Some works reported that a strong basal texture hinders the formability of Mg alloys. Nevertheless, the smaller grain size of the ZX11-350 condition can induce the activation of non-basal slip mechanisms, limiting twinning [60,61,62] and enhancing ductility [22,63,64].

Besides, given the larger grain size and weaker basal texture for the ZX11-450 sample, it is likely to expect a more critical contribution of twin boundaries formation as a deformation mechanism. Additionally, it is seen that fine Mg-Ca precipitates are located along grain boundaries, see Figure 1d. Therefore, it is suspected that the ductility of the ZX11-450 is affected by this microstructural feature. In consequence, the elongation at failure of the ZX11-350 is higher than the ZX11-450 sample in pristine condition, Figure 6a.

A higher annealing temperature reduces the yield strength (YS) since it allows the material to release the strain stored during rolling. According to the Hall–Petch equation, a smaller grain size increases the yield strength. In addition, there is a critical texture effect. It has been reported that a weaker texture induces lower yield stress [65]. Consequently, the YS of the ZX11-450 sample is slightly lower if we compare it with ZX11-350 before immersion, Figure 6c. However, no major contractions for the yield strength (YS) are presented at 15, 30, and 45 days after immersion, for both conditions, in comparison with the pristine condition.

In summary, the most noticeable effect of the annealing treatments on the mechanical properties is observed for the elongation, Appendix A. The higher annealing temperature releases the stored strain, induces grain growth, and the precipitate location at grain boundaries, which negatively induces the fragilization of ZX11-450. Thus, the ZX11-350 sample shows a higher initial elongation than ZX11-450, maintaining this tendency within the testing duration.

The load-bearing capabilities of the Mg alloys are projected to be reduced as the degradation process continues. Thus, for day 15th, there is an elongation reduction of 16% and 40% for ZX11-350 and 450, respectively, compared to pristine conditions. To find a possible reason for the more substantial decrease in the elongation of the ZX11-450 samples compared to the ZX11-350 samples, a microstructural characterization near the fracture surface after 15 days of immersion is presented in Figure 9a,b, respectively. In agreement with the mechanism proposed above, it is possible to observe a fragilization of grain boundaries in both annealing conditions. However, the nucleation of cracks along twins is another clear failure mechanism found in the ZX11-450 samples. This could be an additional reason for the more noticeable decrease in the toughness of the ZX11-450 samples.

Moreover, the lower DR for the ZX11-350 compared to the ZX11-450 sample at the initial stages of immersion could prevent other crack nucleation sites, such as pits. Furthermore, the reduced dissolution of the Mg matrix implies a lower rate of hydrogen evolution. Thus, the risk of hydrogen embrittlement can be diminished [66], also supporting reducing the rate of mechanical integrity loss for the ZX11-350 condition. Nevertheless, both conditions show good stability for longer immersion times.

The potential use of a material in the medical field is determined by extensive in vitro and in vivo evaluation. Usually, there is a mismatch between the performance observed in vitro and in vivo studies, and this is because reproducing the complex physiological conditions in in vitro experiments is arduous. Thus, the degradation rate of Mg alloys implants is higher than in vivo by a factor between 1 and 4 [67]. Consequently, an implant produced by the ZX11 alloy may be expected to present better mechanical integrity in actual life applications. Therefore, it is reasonable to consider that the results presented in this work suggest the feasibility of the ZX11 alloy for its use in orthopedical applications. In particular, the ZX11-350 condition is more appropriate for implants used in some lower limb fractures (toes, calcaneus) since these require good mechanical integrity and a low degradation rate for 6–10 weeks. Meanwhile, the ZX11-450 condition may be more suitable for upper limb fractures (finger, distal radius), where the service period is only 4 to 8 weeks.

### 4.3. Biological Response

MSC cultured in the alloys lixiviates expressed collagen I, Runx-2, osteopontin, and osteocalcin, while the control did not express either Runx-2 or osteocalcin and expressed osteopontin and collagen I. Collagen I expression was similar, with no significant differences between MSC cultured with the alloys lixiviates and the Ctrl; thus, this protein cannot be used to indicate osteogenic differentiation. In the case of osteocalcin, Runx-2, and osteopontin expression, it was significantly (*p* > 0.05) higher for MSC cultured with the alloys lixiviates than for the Ctrl, exhibiting a positive MSC differentiation towards mature osteoblast induced by the alloys. It has been reported that the presence of Mg leads to reduced MSC proliferation and viability (Appendix A shows the decreased viability with culture time for the cells exposed directly to the Mg samples hanging in the inserts). However, it stimulates differentiation towards the osteoblastic phenotype, mainly due to the presence of magnesium in the medium and no other changes due to magnesium corrosion, such as increased pH or osmolality [49,68]. Induction towards osteoblastic differentiation due to MgCl_2_ presence has been linked to activation of the p38/Runx2/Osx signaling pathway [68], in concordance with the present results exhibiting an upregulated expression of Runx-2 for MSC cultured with alloys lixiviates. The upregulated expression of osteocalcin and osteopontin observed for MSC cultured with the ZX11-350 and ZX11-450 lixiviates demonstrated the osteoinductive properties of these alloys. It suggests that a proper formation of matrix bone and bone growth can be expected upon implantation of ZX11-350 or ZX11-450, which in synergy with the biodegradation of the alloys might result in appropriate stability of implants developed from ZX11-350 and ZX11-450 conditions [69].

## 5. Conclusions

The effect of the annealing treatment of the ZX11 alloy on the mechanical integrity, corrosive behavior, and biocompatibility was studied. Two annealing temperatures, 350 and 450 °C were considered. It was shown that the annealing-quenching treatment induces fewer precipitates, smaller grain sizes, and a stronger basal texture for the ZX11-350 condition than for the ZX11-450 condition. The correlation between the microstructural features above-mentioned and the degradation rate and mechanical integrity is summarized as follows:No effect is observed for the short-term (4.5 h) degradation rate determined with electrochemical and hydrogen evolution measurements because competing factors determine the DR.The weak basal texture allows basal <a> slip as an important deformation mechanism. In addition, the smaller grain size induces a higher elongation at failure for the ZX11-350 compared to the ZX11-450 under pristine conditions. Moreover, it is hypothesized that the precipitates at grain boundaries block the dislocations and promote intergranular corrosion leading to the material’s fragilization. In addition to that, nucleation of cracks along twins has been observed to lower elongation to fracture in the ZX11-450 samples.The precipitates located at grain boundaries serve as corrosion nucleation pits, which, combined with a weaker basal texture and larger grain size, results in an increased long-term degradation rate at 7 and 15 days of immersion for the ZX11-450 condition, in comparison with the ZX11-350. Nevertheless, this effect is not observed for more extended periods.The increased degradation rate of the ZX11-450 induces a higher reduction of the elongation at failure and a slight reduction in the yield strength after 15 days of immersion compared to the ZX11-350 condition. However, no further changes were observed afterward.A higher release of Mg ions is observed for the ZX11-350 than for the ZX11-450. Nevertheless, no significant cytotoxic effects were observed.Both annealing conditions revealed osteoinductive properties and enhanced antibacterial activity in *S. aureus* and *S. epidermidis* strains without significant differences.

In accordance with the above-mentioned results, both conditions showed biodegradation synergy with the mechanical integrity evolution, biocompatibility, osteoinductive properties, and antibacterial activity. However, the ZX11-350 condition might be more appropriate for fracture sites that require a higher degree of mechanical integrity stability during the first days after implantation. Meanwhile, the ZX11-450 condition is recommended for implantation sites whose load-bearing demands are lower.

## Figures and Tables

**Figure 1 materials-15-03142-f001:**
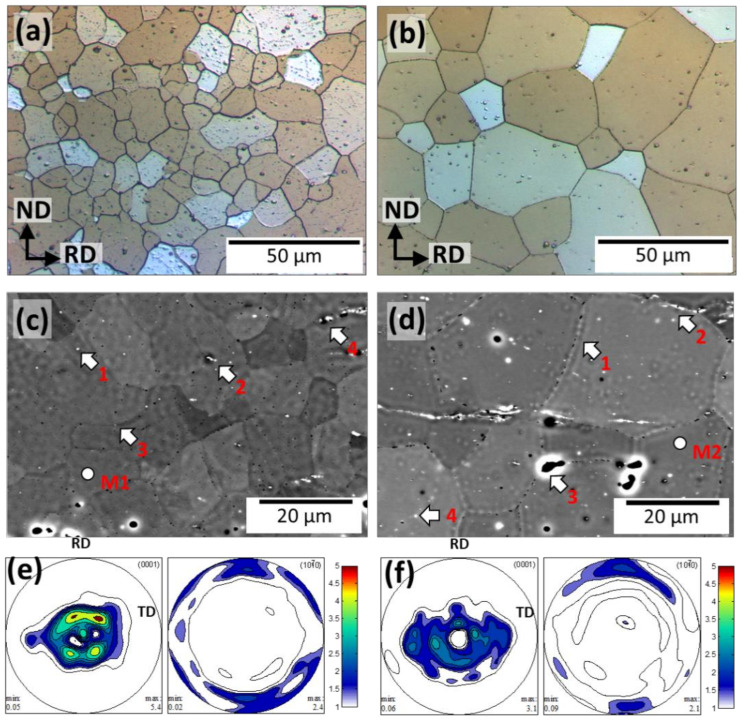
Optical microstructures, phase contrast QBSD-SEM microstructures and crystallographic textures in terms of the $\left(0001\right)$, $\left\{10\bar{1}0\right\}$ pole figures for ZX-350: (**a**,**c**,**e**), and ZX-450: (**b**,**d**,**f**), respectively. Arrows and numbers indicate the zones where the composition was quantified.

**Figure 2 materials-15-03142-f002:**
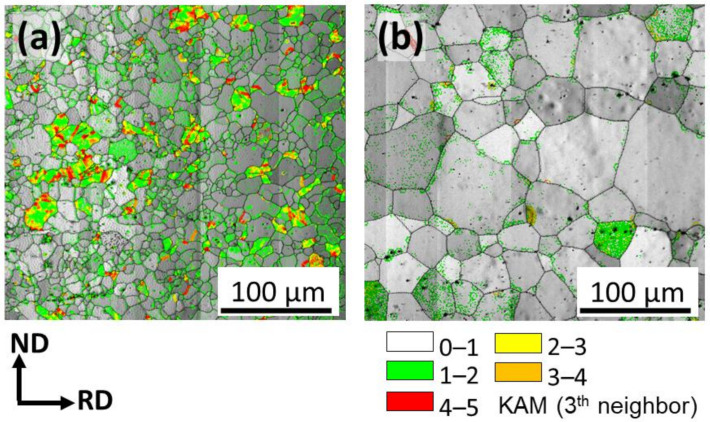
Microstructure of (**a**) ZX11-350 and (**b**) 450 samples using the kernel average misorientation (KAM).

**Figure 3 materials-15-03142-f003:**
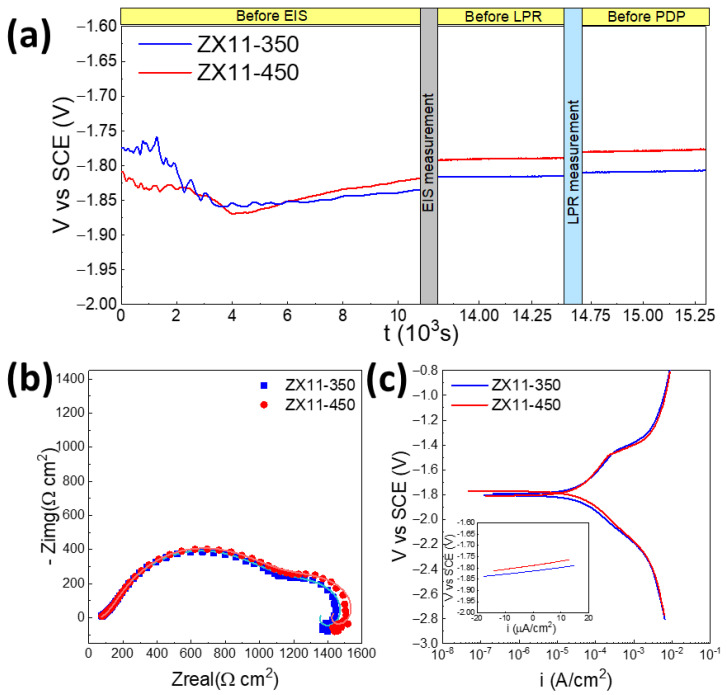
Representative OCP (**a**), Nyquist (**b**) and PDP plots (**c**) of ZX11-350 and ZX11-450 samples immersed in DMEM /F-12 + 10% *v*/*v* FBS. The symbols in (**b**) correspond to the experimental data and the lines to the fitting for the EIS data. All the experiments were conducted in triplicate, observing good reproducibility.

**Figure 4 materials-15-03142-f004:**
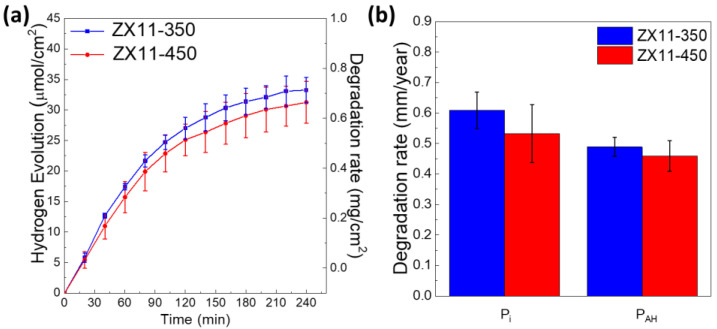
(**a**) Hydrogen evolution of ZX11-350 and ZX11-450 in DMEM/F-12 with 10% *v*/*v* FBS. (**b**) Degradation rate calculated from hydrogen evolution and PDP measurements.

**Figure 5 materials-15-03142-f005:**
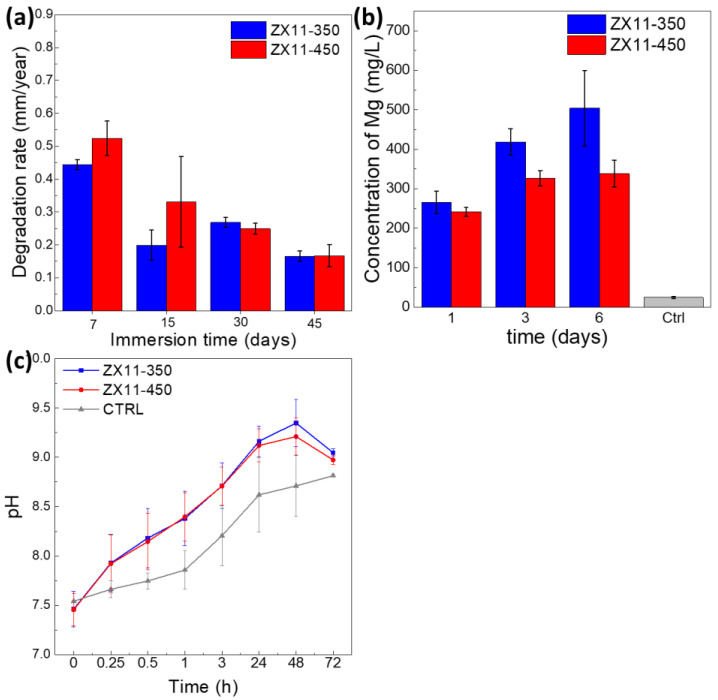
(**a**) Degradation rate from weight loss up to 45 days, (**b**) Concentration of Mg released of ZX11-350 and-450 conditions during immersion in Mg, and (**c**) Plot of pH evolution of the medium during immersion. All measurements were conducted in DMEM/F-12.supplemented with 10% *v*/*v* DMEM/F-12 supplemented with 10% *v*/*v* FBS.

**Figure 6 materials-15-03142-f006:**
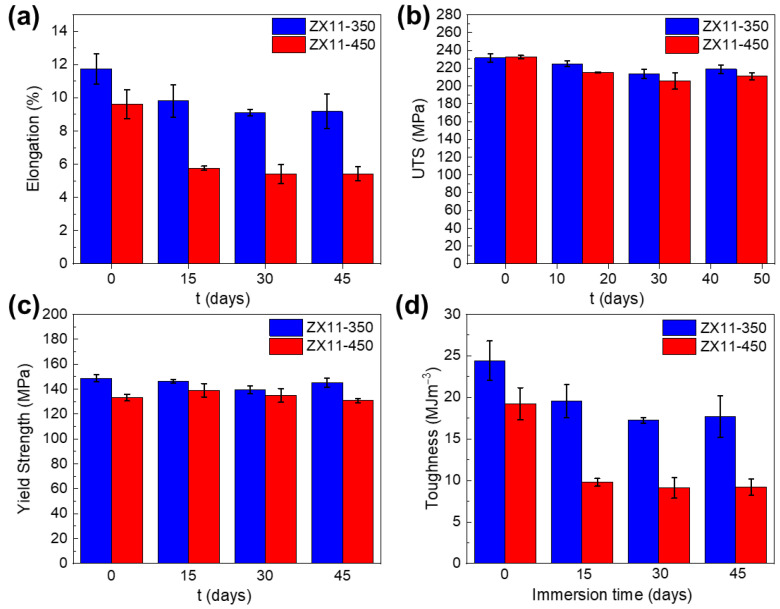
Tensile test experimental results for (**a**) elongation, (**b**) ultimate tensile strength (UTS) (**c**) yield strength, and (**d**) toughness as a function of immersion time.

**Figure 7 materials-15-03142-f007:**
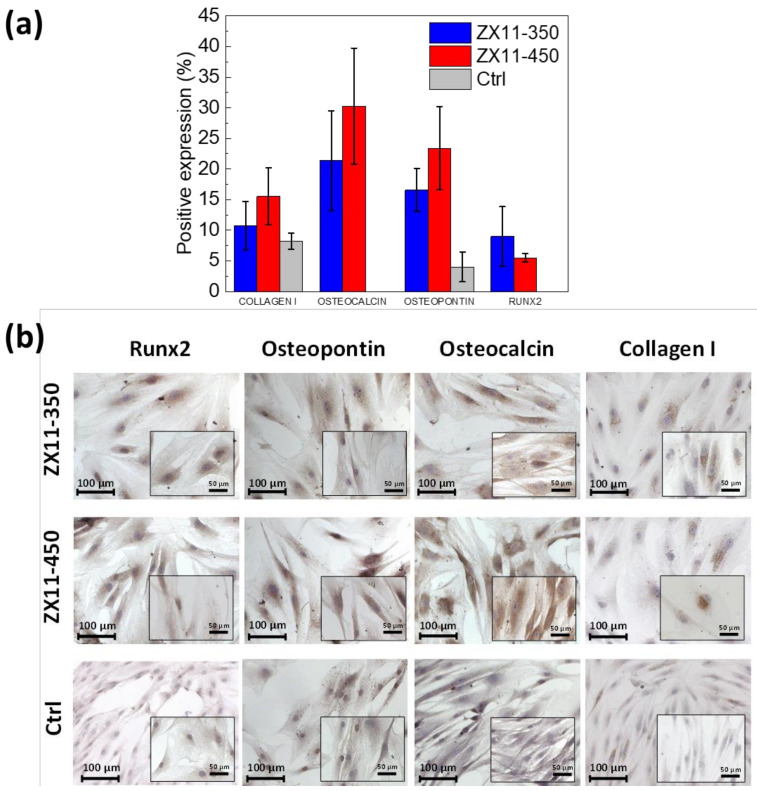
(**a**) Semi-quantitative evaluation, from immunocytochemistry micrographs, of Runx-2, osteocalcin, osteopontin, and collagen I cell expression (**b**) Representative immunocytochemistry micrographs of the expression of Runx-2, osteopontin, osteocalcin, and collagen I, after 8 days of MSC culture in contact with the non-diluted lixiviates from ZX11-350 and ZX11-450. Cells are observed in purple, and positive expression of markers is observed in brown. The scale bar in main micrographs corresponds to 100 µm, while scale bar in inset micrographs corresponds to 50 µm.

**Figure 8 materials-15-03142-f008:**
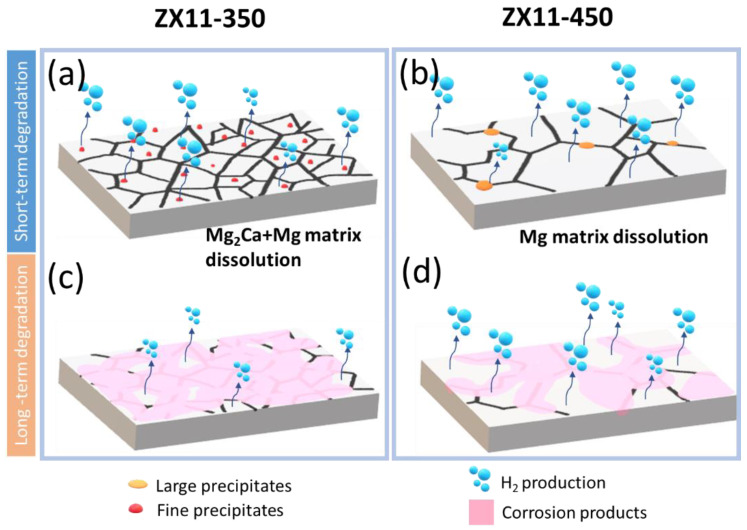
Illustration of the surface activity and passivation for short-term (**a**,**b**) and long-term (**c**,**d**) degradation process of ZX11-350 and ZX11-450 samples.

**Figure 9 materials-15-03142-f009:**
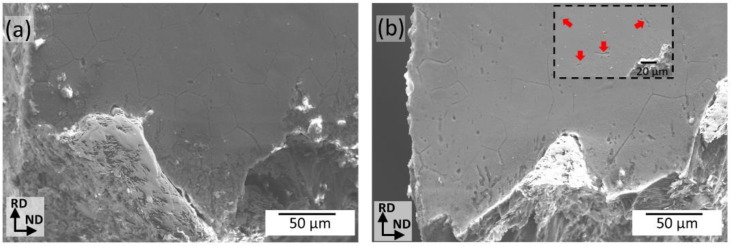
SEM micrographs near the fracture surface of (**a**) ZX11-350 and (**b**) ZX11-450 samples tested in tension after 15 days of immersion. Inset in (**b**) shows a typical crack nucleation along twins in the ZX11-450 samples.

**Table 1 materials-15-03142-t001:** Chemical composition of selected precipitates in Figure 1c,d was determined using EDS.

	Precipitate	Zn	Ca	Fe	Si	O	Mg
ZX11-350	M1	0.63	0.25	-	-	-	Bal.
	1	0.68	0.48	1.8	0.34	-	Bal.
	2	0.67	0.63	-	0.97	-	Bal.
	3	0.58	0.20	-	0.04	-	Bal.
	4	0.67	0.46	0.03	0.07	-	Bal.
ZX11-450	M2	0.51	0.31	-	-	-	Bal.
	1	0.50	0.43	-	0.07	0.60	Bal.
	2	0.54	10.50	-	1.63	1.19	Bal.
	3	0.82	4.02	-	-	0.46	Bal.
	4	0.72	1.68	0.02	0.05	0.60	Bal.

**Table 2 materials-15-03142-t002:** Corrosion evaluation by potentiodynamic polarization and hydrogen evolution (HE).

Alloy	HE	i_corr_ (µA/cm^2^)	P_i_ (mm/y)	Electrolyte	Ref.
ZX11-350	0.48 ±0.06 mm/y	27.32 ± 2.6	0.60 ±0.06	DMEM+FBS	-
ZX11-450	0.45 ± 0.08 mm/y	23.9 ±4.2	0.53 ± 0.09	DMEM+FBS	-
Mg-1Zn-1Ca	0.75 mL/cm^2^	27.37	-	α-MEM	[32]
Mg-3Zn-0.4Ca	1.5 mL/cm^2^	30.25	-	α -MEM	[32]
Mg-2Zn-0.24Ca	~5 mL	21.8	-	SBF	[33]
Mg-1.4Zn-0.5Ca	-	4.28	-	SBF	[34]
Mg-2Zn-06Ca	2.17 mm/y	1.89	-	SBF	[19]
Mg-2Zn-0.2Ca	-	0.332	-	SBF	[35]
Mg–5Zn–1Ca	-	12.16	0.28	SBF	[36]
Mg-5.6Zn-0.7Ca	8 mm/y	17.99	0.41	3.5 wt% NaCl	[31]
Mg-3Zn-0.2Ca	2 mL/cm^2^	4.56	0.1	0.9 wt% NaCl	[37]
Mg-0.9Zn-0.26Ca	~0.5 mL/cm^2^	3.11	0.07	1 mol/L NaCl	[38]
Mg-0.5Zn-0.2Ca	-	31.62	-	3.5 wt% NaCl	[39]

**Table 3 materials-15-03142-t003:** Antibacterial effect (inhibition ratio percentage) of the ZX11 alloy against *E. coli*, *S. aureus*, and *S. epidermidis*. Data are presented as mean ± standard error of the mean (SEM).

	Inhibition Ratio (%)					
	Day 1		Day 3		Day 7	
	ZX11-350	ZX11-450	ZX11-350	ZX11-450	ZX11-350	ZX11-450
*E. coli*	47.5 ± 13.3	56.6 ± 13.2	93.02 ± 2.3	77.6 ± 4.3	51.9 ± 28.4	100
*S. aureus*	99.9 ± 0.003	99.9 ± 0.004	100	100	100	100
*S. epidermidis*	99.9 ± 0.002	99.9 ± 0.01	99.1 ± 0.06	100	99.5 ± 0.06	100

## Data Availability

The data presented in this study are available on request from the corresponding author. The data are not publicly available due to ongoing research.

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
