# Peer review of "Degradation Behavior and Mechanical Integrity of a Mg-0.7Zn-0.6Ca (wt.%) Alloy: Effect of Grain Sizes and Crystallographic Texture"

_materials, 2022, doi:10.3390/ma15093142_

Round 1
Reviewer 1 Report
See the attachment.

Author Response
We (on behalf of all coauthors) would like to thank you for the careful revision of our manuscript and for the valuable comments that allow us to improve the manuscript. Please find attached below a point-by-point answer to your comments. Furthermore, in the manuscript, you will also find additional corrections regarding spelling/writing. All the modifications are indicated by the “track of changes” function of MS word.
Reviewer 1
This is a good paper dealing with the corrosion behavior of a dilute magnesium alloy. I recommend it for publication after minor revision:
The corrosion rate and mechanical properties should be compared with those in recent papers for the Mg-Zn-Ca based alloys, such as Surface and Coatings Technology, 437 (2022) 128328, Rare Metals 40, 1924-1931 (2021), Materials today: Proceedings, https://doi.org/10.1016/j.matpr.2022.02.290, Rare Metals 40, 643-650 (2021), Journal of Magnesium and Alloys, https://doi.org/10.1016/j.jma.2021.12.017, and etc.
Answer: we thank the reviewer for the comment and regarding the suggestion, Table 2 is added in page 11 for comparison with other MgZnCa alloys with similar composition in wt%. Some of the references suggested were also considered.
Reviewer 2 Report
The paper is very interesting, is well written and discuss properlly the effect of the annealing treatment of the ZX11 alloy on the mechanical properties and biocompatibility. In addition to that, this manuscript showed that the release of Mg from both alloys (ZX11350 and ZX11450) induces osteogenesis that is crucial for new bone regeneration and reconstruction.
I just have one question:
Do the authors analyze the gene expression of collagen I, osteocalcin, etc in MSCs by RT-PCR? This tecnhique is helpfull to evaluate the role of Mg2+ in promoting osteogenesis via simulating specific-genes.
Author Response
Dear reviewer,
We (on behalf of all coauthors) would like to thank you for the careful revision of our manuscript and for the valuable comments that allow us to improve the manuscript. Please find attached below a point-by-point answer to your comments.
Reviewer 2
The paper is very interesting, is well written and discuss properly the effect of the annealing treatment of the ZX11 alloy on the mechanical properties and biocompatibility. In addition to that, this manuscript showed that the release of Mg from both alloys (ZX11350 and ZX11450) induces osteogenesis that is crucial for new bone regeneration and reconstruction.
I just have one question:
Do the authors analyze the gene expression of collagen I, osteocalcin, etc in MSCs by RT-PCR? This technique is helpful to evaluate the role of Mg2+ in promoting osteogenesis via simulating specific-genes.
Answer: We thank the reviewer for the positive comment. We agree that a RT-PCR would be an excellent method to study the specific-genes via activated by Mg2+ to promote osteogenesis, which is an interesting topic of study. Yet in the present study, a RT-PCR was not used. For the present manuscript, immunocytochemistry assays, using specific antibodies, to detect the cytoplasmatic presence of proteins characteristic of the osteoblastic phenotype was used as a first approach to identify osteogenesis.
Reviewer 3 Report
Major Points:
- Provide suitable references for the MgZnCa alloys for biomedical applications as discussed in lines 54-56.
- The methods section doesn’t provide references for various techniques. Provide the references for the equations used in the methods. Were there any standards which were used for each kind of testing?
- How many repeats were done for each kind of technique? The text in the results section refers to term 'statistical difference' but there is no mention of which kind of statistical tools were used in the study. Please provide details on the statistics.
- Were the 7-day immersion samples tested for mechanical properties?
Minor Points:
- Provide source for emery paper.
- Figure 7b: The scale bar text is not visible in the images.
Author Response
Dear reviewer,
We (on behalf of all coauthors) would like to thank you for the careful revision of our manuscript and for the valuable comments that allow us to improve the manuscript. Please find attached below a point-by-point answer to your comments.
Reviewer 3
Major Points:
- Provide suitable references for the MgZnCa alloys for biomedical applications as discussed in lines 54-56.
Answer: We appreciate the review and the suggestions of the reviewer. In regard to this major point, references [12,15,16] are provided, in which is reported proper bone healing of fractures using MgZnCa alloys implants.
- The methods section doesn’t provide references for various techniques. Provide the references for the equations used in the methods. Were there any standards which were used for each kind of testing?
Answer: We agree with the comment, therefore the methods section was revised. Lines 166-169 we added to explain the analysis and reference [27] to specify the standard used for electrochemical analysis.
In method section 2.3.2 Hydrogen evolution test lines 183-184 were added. And the performed analysis to determine the number of moles of hydrogen produced by the ZX11 alloy is explained in detail in the supplementary section S2.1.
Reference [28] was added as the standard followed for the mechanical properties evaluation.
- How many repeats were done for each kind of technique? The text in the results section refers to term 'statistical difference' but there is no mention of which kind of statistical tools were used in the study. Please provide details on the statistics.
Answer: We thank the reviewer for this important observation, which has been clarified in the revised manuscript. Details on the statistic analysis used to stablish significant difference for the in-vitro degradation, differentiation and cytotoxicity assays has been presented in the manuscript (section 2.6; statistical analysis). In addition, in methods section it was specified for each technique how many samples were considered.
- Were the 7-day immersion samples tested for mechanical properties?
Answer: According to a previous work (Ref1), which suggested that the evolution of the mechanical properties is not statistically significant for periods between 0 and 15 days. We have made the decision to exclude the mechanical property analysis at 7-day immersion time. We also consider that the conclusions are not affected by this decision.
Ref1: Hou, R.; Victoria-Hernandez, J.; Jiang, P.; Willumeit-Römer, R.; Luthringer-Feyerabend, B.; Yi, S.; Letzig, D.; Feyerabend, F. In Vitro Evaluation of the ZX11 Magnesium Alloy as Potential Bone Plate: Degradability and Mechanical Integrity. Acta Biomater. 2019, 97, 608–622, doi:10.1016/j.actbio.2019.07.053.
Minor Points:
- Provide source for emery paper.
Answer: in the revised manuscript we have specified that SiC emery paper was used for grinding. The producer is also mentioned in the manuscript in page 3, line 127.
- Figure 7b: The scale bar text is not visible in the images.
Answer: We thank the reviewer for this observation. A proper scale bar was added for each main micrograph and the figure caption has been corrected to specifically state the scale bar (in µm) corresponding to the each micrograph group.
Round 2
Reviewer 3 Report
The authors have addressed to the reviewer's comments.